

# Vertical aerosol distribution in the Southern hemispheric Midlatitudes as observed with lidar at Punta Arenas, Chile (53.2°S and 70.9°W) during ALPACA.

Andreas Foth[1,2], Thomas Kanitz[2,a], Ronny Engelmann[2], Holger Baars[2], Martin Radenz[2], Patric Seifert[2], Boris Barja[3], Heike Kalesse[1], and Albert Ansmann[2]

[1]Leipzig Institute of Meteorology, University of Leipzig, Germany
[2]Leibniz Institute for Tropospheric Research, Leipzig, Germany
[3]Atmospheric Research Laboratory, Magallanes University, Punta Arenas, Chile
[a]now at: European Space Agency, ESTEC, Noordwijk, the Netherlands

**Correspondence:** Andreas Foth (andreas.foth@uni-leipzig.de)

**Abstract.** Within this publication, lidar observations of the vertical aerosol distribution above Punta Arenas, Chile (53.2°S and 70.9°W) which have been performed with the Raman lidar Polly$^{XT}$ from December 2009 to April 2010 are presented. Pristine marine aerosol conditions related to the prevailing westerly circulation dominated the measurements. Lofted aerosol layers could only be observed eight times during the whole measurement period. Two case studies are presented showing

long-range transport of smoke from biomass burning in Australia and regionally transported dust from the Patagonian Desert, respectively. The aerosol sources are identified by trajectory analyses with HYSPLIT and FLEXPART. However, seven of the eight analysed cases with lofted layers show an aerosol optical thickness of less than 0.05. From the lidar observations a mean planetary boundary layer (PBL) top height of $1150 \pm 350$ m was determined. An analysis of particle backscatter coefficients confirms that the majority of the aerosol is attributed to the PBL while the free troposphere is characterized by a

very low background aerosol concentration. The ground-based lidar observations at 532 and 1064 nm are supplemented by the AERONET Sun photometers and the space-borne lidar CALIOP on board of CALIPSO. The averaged AOT determined by CALIOP was $0.02 \pm 0.01$ at Punta Arenas from 2009 to 2010.

## 1 Introduction

Aerosols might possibly compensate the warming effect of the greenhouse gases in the Earth's radiation budget within the

uncertainties of future climate modelling (Boucher et al., 2013). The reason for the high uncertainties in the determination of the general aerosol radiative effect is the aerosols variability in their global occurrence, their radiative properties (size, surface, chemistry), and their effects on cloud microphysics.

Global observations with spaceborne sensors improved the understanding of the seasonal distribution of aerosol layers worldwide, e.g. the seasonal vertical distribution of dust as observed with the Cloud-Aerosol lidar with Orthogonal Polarization

(CALIOP,Winker et al., 2007; Liu et al., 2008). Nevertheless, extended field campaigns in key environments of the Earth with homogeneous aerosol conditions provided more-detailed information about properties and cloud-interaction of certain aerosol



types with multi-sensor approaches, e. g. the Tropospheric Aerosol Radiative Forcing Observational Experiment (TARFOX, Russell et al., 1999), the Second Aerosol Characterization Experiment (ACE2, Raes et al., 2000), the Saharan Mineral Dust Experiment (SAMUM, Ansmann et al., 2011) or the Saharan Aerosol Long-Range Transport and Aerosol-Cloud-Interaction Experiment (SALTRACE, Weinzierl et al., 2017).

Within this publication, lidar observations of the vertical aerosol distribution above Punta Arenas, Chile (53.2°S and 70.9°W) are presented as performed during the Aerosol Lidar measurement̂ at Punta Arenas in the frame of Chilean germAn cooperation (ALPACA) campaign which took place from December 2009 to April 2010. This location at the southern tip of South America yields an excellent opportunity to study almost clean marine aerosol conditions which are characteristic for the Southern Oceans (SO), because of the absence of continental land masses in the latitudinal belt south of 45°S and a constant westerly

air flow from the Pacific Ocean (Schneider et al., 2003). The nearest land mass situated towards the prevailing westerlies is New Zealand at a distance of roughly 8000 km and 10° latitude further north. Thus, reported aerosol sources at lower latitudes, like the Amazon rain forest (Reid et al., 2004; Baars et al., 2012), the Patagonian desert (Gaiero et al., 2003; Gassó and Stein, 2007; Li et al., 2010; Johnson et al., 2010), and the Australian continent (Edwards et al., 2006) are expected to barely affect the aerosol conditions at Punta Arenas.

In the framework of the BACCHUS (Impact of Biogenic versus Anthropogenic emissions on Clouds and Climate: towards a Holistic UnderStanding) project, Carslaw et al. (2017) performed simulations to estimate the aerosol conditions in the year 1750 and their impact on climate. It was found that Punta Arenas is in a region that is still representative for pre-industrial aerosol conditions. A similar notation was already previously reported by Hamilton et al. (2014).

These pristine conditions already motivated the ground-based Aerosol Characterization Experiment (ACE1) in the 1990's

(Bates et al., 1998) and were confirmed by Minikin et al. (2003) who contrasted upper-tropospheric in-situ aerosol aircraft observations in the Northern midlatitudes (Scotland) and Southern midlatitudes (Punta Arenas). However, almost no ground-based aerosol and cloud layer profiling was performed in the southern midlatitudes in the following decades, although such measurements would have served as good opportunity to contrast the aerosol radiative effect in the northern and southern midlatitudes with respect to the aerosol sources as well as the influence of aerosols on cloud microphysics (Kanitz et al., 2011;

Kanitz et al., 2013).

In the northern midlatitudes, lidar networks like the European Aerosol Lidar Network (EARLINET) (Bösenberg et al., 2000) monitor aerosol and cloud conditions since almost 20 years (Mattis et al., 2008; Seifert et al., 2010). A comparable network in Latin America, the Latin American Lidar Network (LALINET) was only fully established in 2013 (Guerrero-Rascado et al., 2016; Antuña-Marrero et al., 2017). Moreover, the South American Environment Risk Network (SAVER.Net) was established

by means of a collaboration between Chile, Argentina and Japan to monitor aerosol, ozone, and UV-radiation since 2012 (Ristori et al., 2018). The Atmospheric Research Laboratory of the Magallanes University in Punta Arenas is participating in this activity with a multiwavelength Raman and polarization lidar̂ since 2016 (Barja et al., 2018). As such, the conducted lidar measurements during ALAPACA achieved the most comprehensive data set on aerosol and cloud distribution from ground before the establishment of these networks. The results of ALPACA are presented in this paper. Section 2 gives an overview

about the experiment. The measurement systems used for this study are introduced in Sec. 3. Within ALPACA, lofted aerosol



layers were observed only eight times. Two of these rare cases, including layers of Australian biomass–burning smoke and Patagonian desert dust layers are presented in Sec. 4, followed by an overview of the vertical aerosol distribution in Sec. 5. A conclusion about the results of ALPACA and an outlook about an upcoming campaign are given at the end.

## 2 Experiment

Between the regular shipborne lidar measurements aboard the research vessel Polarstern (Kanitz et al., 2013) in autumn 2009 and spring 2010, the portable lidar Polly$^{XT}$ (Althausen et al., 2009; Engelmann et al., 2016) was deployed at the Magallanes University in Punta Arenas, Chile (53.2°S and 70.9°W) and continuous lidar measurements (24 hours / 7 days a week) were conducted from 4 December 2009 to 4 April 2010, covering a period of four months. Parts of these Polly$^{XT}$ observations have already been evaluated and published by Baars et al. (2016) in the frame of PollyNet.

Figure 1 shows the location of Punta Arenas (red star) in the very south of Chile, at the Strait of Magallanes and between the Pacific and Atlantic Ocean. In this area, the polar front causes a continuous zonal wind band, because of the limited friction of the large ocean surface and the missing structured land masses in contrast to the northern midlatitudes (Cerveny, 1998). Hence, westerly winds prevail the whole year in southern Latin America. Based on the marine environment, daily and seasonal variations of the weather are weak (Coronato and Bisigato, 1998). The annual mean temperature is about 6 °C and the annual
precipitation amounts to 375 mm. Furthermore, the cyclone passage frequency is very high (3-5 days, Hodges et al., 2011). This Southern Ocean region is characterized by a high cloud fraction (> 80 %, Naud et al., 2014) with a cloud fraction of clouds below 3 km of about 60 % (Haynes et al., 2011). The vegetation is composed by grasslands, tundra, and mixed forest.

## 3 Instruments

### 3.1 PollyXT

In the framework of ALPACA the lidar measurements were conducted with the portable multiwavelength Raman and polarization lidar Polly$^{XT}$_IFT (Althausen et al., 2009), as part of the Polly lidar family (Engelmann et al., 2016) and will be referred to Polly$^{XT}$ in the paper. Technically, the lidar is capable to measure the backscattered light at 355, 532, and 1064 nm wavelength, and Raman scattered light at 387 and 607 nm to determine profiles of the particle backscatter coefficient at three wavelengths and the extinction coefficient at 355 and 532 nm. At heights below 400 m the overlap of the laser beam with the receiver field
of view of the bistatic system is incomplete. As a consequence, values of the particle backscatter coefficient were set constant below 400 m height, in the lower part of the planetary boundary layer (PBL) under the assumption of well–mixed conditions. The PBL top height is determined with the wavelet–covariance–transformation that supposes a much higher aerosol load in the PBL than in the free troposphere (Brooks, 2003; Baars et al., 2008).

The rather low aerosol content in the area of Punta Arenas caused low signal–to–noise ratios in the UV and the Raman signals.
Thus, the particle extinction coefficient had to be estimated from the 532 nm and 1064 nm backscatter coefficient by means of appropriate particle lidar ratio ($S_P$) values.



## 3.2 Cloud Aerosol Lidar with orthogonal Polarization (CALIOP)

In April 2006, the Cloud Aerosol Lidar Infrared Pathfinder Satellite Observations (CALIPSO) mission started (Winker et al., 2009). Aboard CALIPSO, the two–wavelength backscatter and polarization lidar CALIOP has been operated to achieve a worldwide four dimensional data set of clouds and aerosols. CALIPSO orbits the Earth in a height of nearly 705 km with a

velocity of $7\,\mathrm{km\,s^{-1}}$ and overpasses the same location every 16[th] day.

The CALIPSO data processing provides profiles of backscatter and extinction coefficient at 532 and 1064 nm within the CALIOP level 2 version 4.10 data. In contrast to level 3 data, the version 4.10 data analysis algorithm aims on distinguishing not six but seven tropospheric aerosol subtypes (Omar et al., 2009, https://www-calipso.larc.nasa.gov/resources/calipso_users_guide/qs/cal_lid_l2_all_v4-10.php). In this study, the CALIPSO data is used to determine planetary boundary layer heights, the

backscatter-related Ångström exponent, and to retrace the long-range transport of smoke based on total attenuated backscatter profiles at 532 nm (level 1 V4.10) and the aerosol subtype product (level 2 V4.10). Level 3 products of the monthly averaged AOT are still based on version 3.10 because version 4.10 is not yet available.

The CALIOP data processing provides several quality flags. Within the scene classification algorithm, the cloud–aerosol–discrimination (CAD) score is determined (Liu et al., 2009). Aerosol particles are assigned with negative values of $-100$ to

$-1$ and clouds are assigned with values of 1 to 100. The larger or smaller the value the more confident is the discrimination, respectively. Values around zero define an uncertain discrimination.

## 3.3 Auxiliary data

An AERONET Sun photometer which measures AOT (column-integrated extinction coefficient) from 340 to 1020 nm at 7 channels is located at Rio Gallegos ($51.6°$S, $69.3°$W,), Argentina (CEILAP-RG). Level 2.0 data is used with an AOT uncer-

tainty of 0.01 to 0.02 (Holben et al., 2001).

Two models were used to support the analysis of the air mass transport. HYSPLIT is a model to calculate trajectories of air parcels for simulations of dispersion and deposition at arbitrary locations (Draxler and Hess, 1998; Stein et al., 2015). By means of trajectories, aerosol sources are assigned. FLEXPART is a Lagrangian particle dispersion model, which calculates probabilistic trajectories of a large number of air parcels (Stohl et al., 2005). Thereby, the transport and diffusion of aerosol

could be described and a coarse assignment of aerosols to their sources is enabled.

In the statistical analysis of the aerosol conditions during ALPACA, we used ensemble backward trajectories combined with a land cover classification for a temporally and vertically resolved airmass source attribution. The land cover is a simplified version of the MODIS land cover (Friedl et al., 2002). At first, a 27-member ensemble of 10-day backward trajectories is calculated using HYSPLIT. Meteorological input data for HYSPLIT are taken from the Global Data and Assimilation Service

dataset (GDAS1, https://www.ready.noaa.gov/gdas1.php) provided by the Air Resources Laboratory (ARL) of the U.S. National Weather Service's National Centers for Environmental Prediction (NCEP). Each ensemble is generated using a small spatial offset in the trajectory endpoint. Whenever a trajectory is below the mixing depth provided in the GDAS1 data ('reception height'), the land cover is categorized using custom defined polygons according to land mass boundaries. Hence, an



air parcel is assumed to be influenced by the surface if the trajectory is below the mixing depth. The residence time for each category is then the total time an air parcel fulfilled this criterion by land cover category. This calculation is repeated in steps of 3 h in time and 500 m in height to provide a continuous estimate on the airmass source and as a first hint on potential aerosol load.

The global aerosol model NAAPS (Navy Aerosol Analysis and Prediction System) is used to obtain modelled aerosol optical thickness of dust (https://www.nrlmry.navy.mil/aerosol/).

## 4   Case studies

In this section, two cases are discussed in detail. The first one shows two lofted smoke layers after intercontinental long-range transport and low level Patagonian dust. In the second one, further low-level Patagonian dust was observed.

### 4.1   2 March 2010: Lofted smoke layers and Patagonian dust

On 2 March 2010, the weather at Punta Arenas was dominated by a north–westerly air flow. Between 09:00 UTC and 12:00 UTC, Polly[XT] observed clouds in the range from 4 to 5 km and a PBL height of about 1 km (Fig. 2). An aerosol layer was situated above the PBL up to a height of 2 km. An increased backscatter was also observed at the height range from 4.5 to 5.2 km (06:00 to 12:00 UTC) and from 11.5 to 12.1 km (06:00 to 09:30 UTC). The source apportionment of the three observed
aerosol layers by means of FLEXPART and HYSPLIT analyses, is shown in Fig. 3, 4 and 5. In Figure 3a, a FLEXPART analysis for the lower layer (1 to 2 km) reveals that the air masses passed the southern Pacific and southern Patagonia (red colouration in Fig. 3a). HYSPLIT 48–hours backward trajectories for heights of 0.5, 1 and 2 km confirm that the ground–level aerosol on 2 March 2010 was locally affected by an extended residence time above Patagonia (Fig. 4). At this time, modelled aerosol optical thickness (AOT) from the global aerosol model NAAPS locally exceeded 0.14 and 0.02 in the area of Punta
Arenas (Fig. 4).

The FLEXPART analyses for the layers observed between 4.5 and 5.2 km and 11.5 and 12.1 km height reveal a clearly defined region of origin. The observed layers were advected to Punta Arenas from Australia and the southern Pacific (Fig. 3b and c). HYSPLIT 13–days backward trajectories were calculated as well (blue and light blue in Fig. 5). Figure 5 (top) also shows active fire spots (red dots) derived by MODIS (Moderate Resolution Imaging Spectroradiometer, Justice et al., 2011), which
were detected between 17 and 25 February 2010. The layer between 4.5 and 5.2 km (blue) as well as the layer between 11.5 and 12.1 km (light blue) crossed regions with fire activity in South and Southeast Australia. In summary, the source apportionment study for 2 March 2010 reveals that the surface-near aerosol layer is likely dominated by Patagonian dust, whereas the two observed lofted layers contain long-range transported smoke from Australia.

The Polly[XT] measurement was analysed for the cloud–free period between 06:15 and 09:00 UTC on 2 March 2010. Figure 6
illustrates the profiles of the optical properties with a smoothing length of 150 m. The particle backscatter coefficients reach their maximum values of up to $1.02\,\mathrm{Mm^{-1}sr^{-1}}$ (532 nm) and $0.63\,\mathrm{Mm^{-1}sr^{-1}}$ (1064 nm) in the PBL. The observed layers in the height–time display (Fig. 2) are also slightly visible in the profiles of the particle backscatter coefficient in heights of 5 and



12 km. The profile of the particle extinction coefficient was reproduced for an assumed constant lidar ratio. According to the origin of the air masses, a lidar ratio of $40 \pm 10$ sr (Kanitz et al., 2013) was applied to the ground–level layer (up to 2.5 km) and a lidar ratio of $70 \pm 10$ sr (smoke Ansmann et al., 2009; Tesche et al., 2011) was used for the lofted layers (Fig. 6c). The AOTs of each single layer result in values of $0.044 \pm 0.004$ (0 to 2.5 km), $0.004 \pm 0.0004$ (4.5 to 5.2 km) and $0.002 \pm 0.0002$ (11.5 to

12.1 km). The Ångström exponent (Fig. 6d) amounts to $0.56 \pm 0.21$ in the ground–level layer (up to 2 km). Kanitz et al. (2013) determined a Patagonian–dust–related Ångström exponent of $0.4 \pm 0.1$. In the framework of EARLINET, Saharan–dust–related Ångström exponents of $0.5 \pm 0.5$ were determined (Müller et al., 2007). In the smoke layer between 4 and 5.5 km, the Ångström exponent was $0.61 \pm 0.1$. In comparison, biomass–burning–smoke–related Ångström exponents of $1.0 \pm 0.4$ were determined by lidar observations for long-range transported smoke from Siberia and Canada (Müller et al., 2007). In Cape Verde, Ångström

exponents of smoke originating from the south of western Africa, were $1.06 \pm 0.65$ (Tesche et al., 2011). Smoke that was transported from Africa to the Amazon rain forest was found to have Ångström exponents of 0.8 (Baars, 2012).

   In a next step, CALIPSO lidar observations were used for the characterization of the lofted aerosol layers during long-range transport. Six CALIPSO overpasses were found for the observation of the two lofted layers (see green lines in Fig. 5 top), which provide intersections of the intercontinental transport of the smoke plume. On 21 February 2010, CALIPSO overpassed the

region of origin of the smoke in Southern Australia. Figure 7a and b show the height–time display of the attenuated backscatter coefficient and the determined aerosol subtypes (with CAD score $< -80$) on 21 February 2010. Between heights of 4 and 7 km, a section of increased backscatter can be identified. According to the CALIOP data algorithms, the lofted aerosol layer is a mixture of smoke (black) and continental aerosol (green), which confirms the analysis of their origin discussed above. The AOT of the layer determined by CALIOP amounts to 0.1 (at 532 nm). AERONET Sun photometer measurements show a mean

AOT of 0.165 (at 500 nm) in Canberra, Southern Australia, on 21 February 2010. In contrast, a biomass–burning–related AOT of 0.55 (at 532 nm) was determined in the Amazon rain forest during the dry season (Baars, 2012).

On 22 February 2010, the smoke layer is also visible between heights of 4 and 8 km (Fig. 7c and d). On the following days, CALIOP is not able to determine unambiguously the smoke layer (Fig. 7e,f,g and h). A possible reason might be the decreasing smoke concentration along its transport route caused by dispersion and deposition (Bigg, 1973). The simultaneous occurrence

of clouds and aerosol (on 24 February, see Fig. 7e,f at 8 km height and on 27 February at 5 to 11 km height, see Fig. 7g,h), which constrains the detection of the optically thin smoke plume, might be another reason. However, the CAD score of the detected smoke layers is larger than $-80$ which indicates high discrimination accuracy (Liu et al., 2009). In the region of Punta Arenas (on 2 March 2010 in a height of 4 km), CALIOP is again not able to doubtlessly determine the smoke layer, because all determined aerosol layers are in the vicinity to clouds. On 3 March 2010, the smoke layer was confidently detected by CALIOP

(Fig. 7l).

## 4.2   17 February 2010: Patagonian dust

This subsection discusses the observation of a low-level aerosol layer that was observed with Polly$^{XT}$ on 17 February 2010 (see Fig. 8). A cyclone situated in the northwest of Punta Arenas caused northerly air flows on this and the previous days. The calculated 96-hour HYSPLIT backward trajectories for 0.5, 1, 1.5 and 2 km reveal the South Pacific and southern Patagonia



as origin of the air masses (Fig. 9). The corresponding modelled NAAPS AOT exceeded 0.2 in southern Patagonia during this time (Fig. 9).

In the lidar measurement between 00:00 and 06:00 UTC on 17 February 2010, an enhanced aerosol load can be deduced by an increased backscatter coefficient up to 2.5 km height. The surface layer below 1.2 km height seems to contain two sublayers,

one below and one above 0.6 km height, respectively.

Vertical profiles of the particle backscatter coefficient of this measurement were generated for the time period from 02:00 to 06:00 UTC and a vertical smoothing length of 150 m (framed in Fig. 8 bottom). It is illustrated up to 5 km because the atmosphere was free of clouds and aerosol above this height. Figure 10a shows the particle backscatter coefficients at 532 nm (green) and 1064 nm (red), as derived with the Raman method. Both reach their maximum values of $2.5\,\mathrm{Mm}^{-1}\mathrm{sr}^{-1}$ (532 nm)

and $1.8\,\mathrm{Mm}^{-1}\mathrm{sr}^{-1}$ (1064 nm) in the PBL. In the profiles of the particle backscatter coefficient, an aerosol layer up to 3 km height is visible. At heights above 3 km, the particle backscatter approaches zero. The profile of the particle extinction coefficient (532 nm) was reproduced for a Patagonian-dust-related lidar ratio of $40 \pm 10$ sr (Kanitz et al., 2013) (Fig. 10b). The particle extinction coefficient reaches maximum values of $100\,\mathrm{Mm}^{-1}$. The vertical integration of the particle extinction coefficients below 3 km results in an AOT of $0.09 \pm 0.01$. That AOT values is well between the ones that would be derived for lidar

ratios of other species of dust. For instance, assuming Saudi-Arabian dust (lidar ratio of 38 sr, Müller et al., 2007), Indian dust (43.8 sr, Schuster et al., 2012) or Saharan dust ($53 \pm 7$, Preißler et al., 2011) would yield AOTs of 0.08, 0.1, and 0.12, respectively. Spaceborne measurements with MODIS between 16 and 18 February 2010 however indicate an increased AOT $> 0.2$ over the southern Patagonian desert. In the region of Punta Arenas, the AOT amounts to 0.07 up to 0.1. Whether the increased AOT close to Punta Arenas is caused by dust load could not be clearly identified. Comparable AOTs were also determined by

MODIS in the southwest of Patagonia.

The mean Ångström exponent was $0.48 \pm 0.06$ (Fig. 10d) which is within the range of the values for Patagonian dust described in section 4.1 and which is also confirmed by Müller et al. (2007) and Kanitz et al. (2013).

## 5 Statistical results

### 5.1 General aerosol conditions

The general aerosol conditions in Punta Arenas are presented in Fig. 11a in terms of monthly averaged AOT at 532 nm wavelength as determined with CALIOP for the grid cell of Punta Arenas (blue curve) from 1 January 2009 to 31 December 2010. The averaged AOT at Punta Arenas from 2009 to 2010 was $0.02 \pm 0.01$. The annual course of the monthly averaged AOT indicates the absence of a pronounced seasonal cycle. The large standard deviation may have been caused by the low numbers of observations, but may also be the result of the CALIOP aerosol typing limitations in coastal regions (Kanitz et al., 2014).

For comparison, the AOT as obtained at the AERONET station of Rio Gallegos is shown, too. At Rio Gallegos the mean AOT between 2009 and 2010 ($0.02 \pm 0.02$) is in the same range as at Punta Arenas, although Rio Gallegos is situated closer to the Patagonian desert and 400 km east of the west coast of Latin America. AERONET AOT measurements at Rio Gallegos confirm the very low mean AOT values (2009: $0.02 \pm 0.01$, 2010: $0.02 \pm 0.01$) of CALIOP and indicate clean marine conditions



at Punta Arenas and Rio Gallegos (Fig. 11b). Such low AOT values were also found in other coastal and remote oceanic areas. During three meridional transatlantic cruises from 50°N to 50°S shipborne lidar measurements revealed AOTs of the marine boundary layer of below 0.05 in 78% of the cases (Kanitz et al., 2013). Wilson and Forgan (2002) determined a mean AOT of 0.01 at Cape Grim, Tasmania from 1986–1999. AERONET Sun photometer measurements in marine areas show AOTs of

$0.085 \pm 0.01$ over the Pacific and $0.06 \pm 0.02$ over the Southern Ocean (Smirnov et al., 2009).

Figure 11 (c) shows the height–time display of the range–corrected signal at 1064 nm wavelength measured by Polly[XT] for the entire measurement period. As expected, most of the aerosol load is contained within the PBL up to around 1200 m (reddish colors). The free troposphere is characterized by a very low aerosol load but a frequent occurrence of clouds (grey and white colors).

A comprehensive analysis of aerosol source regions based on ensemble HYSPLIT trajectory calculations shows (Fig. 11d) that the influence of the ocean on air parcels (given in accumulated residence time of the backward trajectories within the PBL over the respective surface type) reaching Punta Arenas is a factor of 100 larger in contrast to the continents Africa, Australia and even South America.

## 15  5.2  Vertical aerosol distribution

After introducing the general aerosol conditions at Punta Arenas, in this section the vertically resolved measurements with Polly[XT] are investigated to determine the vertical aerosol distribution. First of all, the ALPACA lidar measurements are applied to ascertain the height of the PBL. Figure 12a presents the monthly means of the PBL heights. During the period with the strongest warming in the southern hemispheric summer (December and January), the maximum averaged top heights were

reached ($1230 \pm 331$ m and $1177 \pm 365$ m, respectively). The PBL heights decrease to $1106 \pm 317$ m and $984 \pm 347$ m, respectively, in February and March, due to decreasing solar irradiation. However, the trend to low values is within the range of the standard deviation, which is indicated as error bars. Conducting stationary and continuous lidar measurements with Polly[XT], the determination of the temporal development of the PBL heights is possible (Cohn and Angevine, 2000; Baars et al., 2008). Figure 12b shows the averaged diurnal variation in the PBL height with a time resolution of three hours. However, no explicit

diurnal variation is identifiable within the limits of the error bars. The values vary between $850 \pm 290$ m and $1280 \pm 370$ m. In contrast for a northern hemispheric site, PBL heights larger than 2 km were observed in 59 % of all considered cases in Leipzig (51°N, 12°E, Mattis et al., 2008). Figure 13 a illustrates the frequency distribution of the obtained PBL heights as obtained from Polly[XT], CALIOP, radiosonde, and GDAS1. For the ground-based lidar observations, in 74 % of all cases, the PBL extends up to heights between 750 and 1500 m and the mean PBL top height amounts to $1151 \pm 347$ m. CALIPSO overpassed

Punta Arenas 31 times in the period from 1 May 2009 to 30 April 2010. The CALIPSO aerosol subtype dataset is used for the classification of the PBL height. From these, the upper limit of the lowest aerosol layer, if present at all, was assumed to be the PBL height. In total, PBL heights could be esstimated for a total of 21 overpasses. The mean PBL height identified with CALIOP is $1169 \pm 532$ m (Fig. 13b). 52 % of the PBL heights determined by CALIOP are between 750 and 1500 m. Furthermore, PBL heights were derived from radiosondes (each at 12 UTC) at the airport of Punta Arenas (37 m ASL) between 4





December 2009 to 31 March 2010 (Fig. 13). The mean PBL height is $1019 \pm 376$ m and 64 % of all values are between 750 and 1500 m. 73 % of all radiosonde ascents show a further lower layer with a mean height of $190 \pm 13$ m which is indicated by a strong temperature gradient. The PBL thus seems to consist of two thermodynamically distinct layers. The reason for that may be explained by the terrain in the area of Punta Arenas which is directly located at the Strait of Magallanes. The

southern Andes mountains and the Pacific are situated in the west (Fig. 1). The lower aerosol layer might be considered as the inflow of the marine boundary layer into the continental boundary layer. However, this layer is rarely detected by Polly$^{XT}$ due to the effect of incomplete overlap below 400 m height. For comparison, only the upper layer is considered. The PBL heights provided by the GDAS1 data set (see Sec. 3.3) are calculated on the basis of the Gradient Richardson number (Stull, 1988) and their mean is about $1024 \pm 463$ m (Fig. 13). Although, the radiosonde data is assimilated into GDAS1 data set, the peak

in low PBL heights derived from the radiosondes does not appear in the GDAS1 data. The horizontal resolution of GDAS1 data is $1° \times 1°$ and the vertical one is approximately 250 m in the lowermost 1000 m. Locations between these grid points are interpolated which might explain the observed differences in the PBL height distributions. In 6 % of all cases, the GDAS1 PBL heights solely fall below 250 m. Generally, the PBL heights determined by CALIOP, radiosondes and the GDAS1 data are in agreement with the values derived from the Polly$^{XT}$ observations which indicates that the aerosol accumulates mainly

within the PBL in the region of Punta Arenas. In the following, statistics of the vertical aerosol distribution are presented. During the ALPACA campaign, 59 measurement periods were analysed. The low amount of determined aerosol profiles is caused by the frequent occurrence of low and mid-level clouds (about 83 % of the measurement period, Kanitz et al., 2011) hampering an evaluation of the aerosol lidar data. Furthermore, the analysis is limited due to the frequent presence of marginal aerosol concentrations in the atmosphere and a corresponding low signal-to-noise ratio measured in the lidar signals. Figure 14

shows the analysed clear-sky profiles of the particle backscatter coefficient at 532 nm (a) and 1064 nm (b). The general vertical smoothing length was 330 m. Well-mixed homogeneous aerosol conditions are assumed to be present in the PBL, such that the particle backscatter coefficient is set constant in the overlap region below 400 m height. The profiles of the particle backscatter coefficient which were derived by either Raman or Klett method (Fig. 14) indicate, that the particle backscatter coefficient is close to $0$ Mm$^{-1}$sr$^{-1}$ above a height of 2 km. Hence, the majority of the aerosol is located in the PBL, as also indicated in the

good correlation between radiosonde and Polly$^{XT}$ derived PBL heights, discussed above. The particle backscatter coefficients reach mean values of $0.74 \pm 0.56$ Mm$^{-1}$sr$^{-1}$ (532 nm) and $0.43 \pm 0.32$ Mm$^{-1}$sr$^{-1}$ (1064 nm). However, aerosol layers were almost never observed in the free troposphere. Therefore, the free troposphere may be considered as a region representing pristine background aerosol conditions making it an ideal region as low-aerosol reference for studies of aerosol-cloud interactions (Kanitz et al., 2011).

Lidar measurements as well as AERONET measurements provide spectrally resolved information. The spectral behavior is expressed by means of the Ångström exponent. Figure 15 displays the frequency distribution of the vertical integrated backscatter-related Ångström exponent (at 532 and 1064 nm) for the time periods from 4 December 2009 to 4 April 2010 using Polly$^{XT}$ (a) and 1 May 2009 and 30 April 2010 using CALIOP (b). For comparison, Figure 15c illustrates the AOT-related Ångström exponent of the AERONET station in Rio Gallegos for the ALPACA period. In all three cases, the maximum values of the

distribution are between 0 and 1. The mean values are $0.8 \pm 0.3$, $0.3 \pm 0.7$ and $0.5 \pm 0.5$ for the measurements with Polly$^{XT}$,



CALIOP and AERONET, respectively. By means of these values, the size of the particles can be derived qualitatively. Very low or negative Ångström exponents indicate very large aerosol particles (sea salt or dust) (Moulin et al., 1997; Müller et al., 2007). In turn, large Ångström exponents (>2) point towards small aerosol particles (such as fresh smoke, Baars et al., 2012). The fraction of low Ångstöm exponents is largest for Rio Gallegos. This is caused by the occurrence of more dust events in

the outflow of Patagonia. The values derived by CALIOP are too noisy to allow a reasonable interpretation. The Ångstöm exponents determined by Polly$^{XT}$ are representative for continental aerosol as the lowermost heights of the atmosphere below 400 m height containing most likely a marine contribution could not be analysed due to instrumental limitations as discussed above.

## 6    Conclusions & Outlook

The presented study aimed on providing an overview about the vertical aerosol conditions above Punta Arenas. During the 4 months of observations of Polly$^{XT}$, lofted aerosol layers were rarely observed and, when present, were characterised by very low optical thicknesses. Overall, the mean aerosol optical thickness at Punta Arenas was found to be 0.02, which is very low, even for marine conditions. Our study thus confirms well the conclusions of Hamilton et al. (2014) and Carslaw et al. (2017) that the atmosphere over southern Chile still provides pristine, pre-industrial conditions. CALIPSO observations, which were

utilized to track the long-range transport of aerosol from Australia to Punta Arenas, indicate that a considerable fraction of free-tropospheric aerosol is removed by cloud processes and washout taking place over the Pacific Ocean before it reaches South America. The average free-tropospheric aerosol load is thus subject to increase to the west of Punta Arenas. The same can be expected eastward of South America, because, as was found in analyses of NAAPS aerosol model simulations, sub-stantial amounts of Patagonian dust are frequently emitted into the atmosphere and transported lee-ward of South America.

We thus conclude from our study that Punta Arenas is one of the few accessible places on Earth with temperate climate where aerosol-cloud-interaction reference studies in the absence of free-tropospheric aerosols can be conducted. Nevertheless, the significance of the ALPACA dataset is somewhat limited because the lidar observations suffered from a rather large height of full overlap and the absence of polarization and UV measurements. These conditions hampered us from more detailed analyses of the aerosol type and the marine contribution to the total aerosol load. As a consequence, the Polly$^{XT}$_IFT system was mean-

while reconstructed to improve the performance under very different aerosol conditions. It is now equipped with a better data acquisition in combination with new photo multiplier tubes, a near-range receiver, and an additional depolarization channel as well as a water vapour detection channel and is named Polly$^{XT}$_TROPOS (Engelmann et al., 2016).

Future studies might enhance the knowledge of the aerosol conditions regarding an aerosol typing (Baars et al., 2017) and separation of aerosol types as well as an estimation of ice nucleating particles and cloud condensation nuclei from multiwave-

length Raman and polarization lidar observations (Mamouri and Ansmann, 2015).

In recent years the Southern Ocean, especially the area in the South and South East of Australia, became an increasing focus for atmospheric researchers. Several observation campaigns like MARCUS (Measurements of Aerosols, Radiation, and Clouds over the Southern Ocean), SOCRATES (the Southern Ocean Clouds Radiation Aerosol Transport Experimental Study)



and CAPRICORN (Clouds, Aerosols, Precipitation Radiation and atmospherIc Composition Over the southeRN ocean, Protat et al., 2016; Mace and Protat, 2018a, b) provide beneficial data for improving the understanding of aerosol-cloud-interaction in pristine environments.

In the upcoming field experiment organized by TROPOS in collaboration with the Magallanes University, Punta Arenas (UMAG) and the Institute for Meteorology at the University of Leipzig (LIM) an extended version of the Leipzig Aerosol and Cloud Remote Observation System (LACROS, Bühl et al., 2013) will be deployed at UMAG in November 2018 for at least one year to study the seasonal cycle of aerosols and clouds in Punta Arenas. LACROS comprises, amongst others, a Polly$^{XT}$ lidar, a 35 GHz cloud radar, a Doppler lidar, a microwave radiometer, a distrometer and radiation sensors for direct and diffuse downwelling and upwelling solar and thermal radiation. During this field experiment named DACAPO-PESO (Dynamics, Aerosol, Cloud and Precipitation Observations in the Pristine Environment of the Southern Ocean), the LACROS suite is extended by a 94 GHz frequency-modulated continuous wave cloud radar provided by LIM (LIMRAD94, Küchler et al., 2017) and a 24 GHz micro rain radar (TROPOS) which will allow for multi-frequency polarimetric Doppler radar studies. Complementing instrumentation of UMAG includes radiosondes, radiation observations, in-situ aerosol observations, and multi-wavelength lidar measurements. The latter are performed since 2015 in the frame of LaLiNet (Guerrero-Rascado et al., 2016; Antuña-Marrero et al., 2017).

With this extended observational suite, several research questions like the efficiency of liquid-dependent ice formation as well as the influence of aerosol concentration on the frequency of occurrence of mixed-phase cloud processes like aggregation and riming will be addressed. These observations might help to improve the representation of Southern Ocean clouds in global climate models which currently suffer from a strong radiation bias (Bodas-Salcedo et al., 2014) caused by a misrepresentation of cloud phase. Specifically, the amount of ice is overestimated by models while observations show a large amount of clouds with high supercooled-liquid water content at cloud top (Haynes et al., 2011). Since cloud thermodynamics are a strong function of cloud condensation nuclei and ice nucleating particles availability which in turn are related to the aerosol load and type (Mamouri and Ansmann, 2015) the results from the ALPACA campaign, presented here, built a fundamental basis of knowledge about the aerosol conditions, cloud condensation nuclei and ice nucleating particles that can be expected at this site.

*Data availability.* The Polly lidar data are available at TROPOS upon request (polly@tropos.de). CALIPSO data were downloaded from the NASA Atmospheric Science Data Center web page (http://www-calipso.larc.nasa.gov/). Backward trajectories analysis was supported by air mass transport computation with the NOAA (National Oceanic and Atmospheric Administration) HYSPLIT (HYbrid Single-Particle Lagrangian Integrated Trajectory) model (HYSPLIT, 2018) using GDAS meteorological data (Stein et al., 2015; Rolph et al., 2017). AERONET sun photometer AOT data were downloaded from the AERONET web page (http://aeronet.gsfc.nasa.gov/). Data from the global aerosol model NAAPS (Navy Aerosol Analysis and Prediction System) were downloaded from the NAAPS web page (https://www.nrlmry.navy.mil/aerosol/).



*Author contributions.* AF prepared the manuscript in close cooperation with TK, HB, PS, BB and HK. AF and TK performed the investigations and data analyses. TK, HB and RE realized the experimental set-up and were responsible for the high-quality of the lidar measurements. RE realized the optical and technical setup of the lidar. HB provided the software for the analysis of the lidar data and supported the data analysis and interpretation. MR developed and applied the code for the determination of the accumulated residence time of backward trajectories separated by different regions of origin. The conceptualization was initialized by AA. All authors have contributed to the scientific discussions.

*Competing interests.* The authors declare that they have no conflict of interest.

*Acknowledgements.* First of all we thank Dietrich Althausen (TROPOS), Felix Zamorano (UMAG) and Claudio Casiccia (UMAG). Without your support the lidar could not have been operated in Punta Arenas.

The authors acknowledge support through the High-Definition Clouds and Precipitation for advancing Climate Prediction research program (HD(CP)[2]; FKZ: 01LK1504C and 01LK1502N) funded by Federal Ministry of Education and Research in Germany (BMBF), ACTRIS under grant agreement no. 262254 of the European Union Seventh Framework Programme (FP7/2007-2013), ACTRIS-2 under grant agreement no. 654109 from the European Union's Horizon 2020 research and innovation programme, EUCAARI funded by the European Union (FP6, grant no. 036 833-2) and the Gottfried Wilhelm Leibniz Association (OCEANET project in the framework of PAKT).

Contributions by Heike Kalesse were made with support of the project PICNICC, GZ: KA 4162/2-1 within the Priority Programme PROM of the German Science Foundation DFG.

We also thank the NASA Langley Research Center and the CALIPSO science team for the constant effort and improvement of the CALIPSO data. Supplementary information from HYSPLIT trajectories, NAAPS aerosol modelling and MODIS was a cornerstone of our data analysis. We especially acknowledge the work of Brent Holben, Eduardo Quel, Lidia Otero, Jacobo Salvador for operating the AERONET station at Rio Gallegos.



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



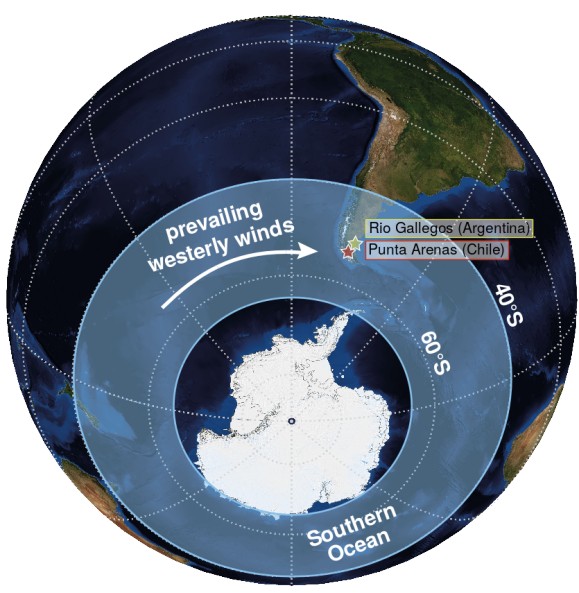

**Figure 1.** Map of Antarctica and South America. Punta Arenas and Rio Gallegos are marked by a red and a yellow star, respectively.

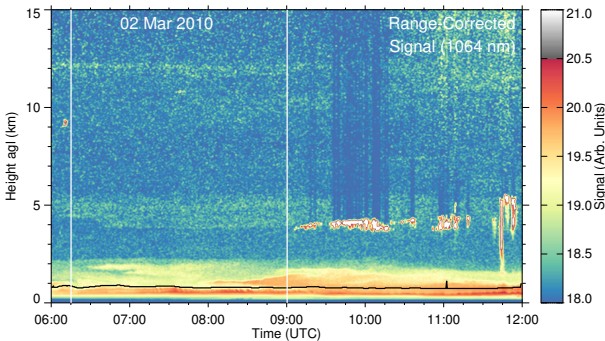

**Figure 2.** Height–time display of the range–corrected signal observed at 1064 nm with the Polly$^{\text{XT}}$ lidar on 2 March 2010. The analysed period between 06:15 bis 09:00 UTC is framed white. The planetary boundary layer top height is indicated by the black line.




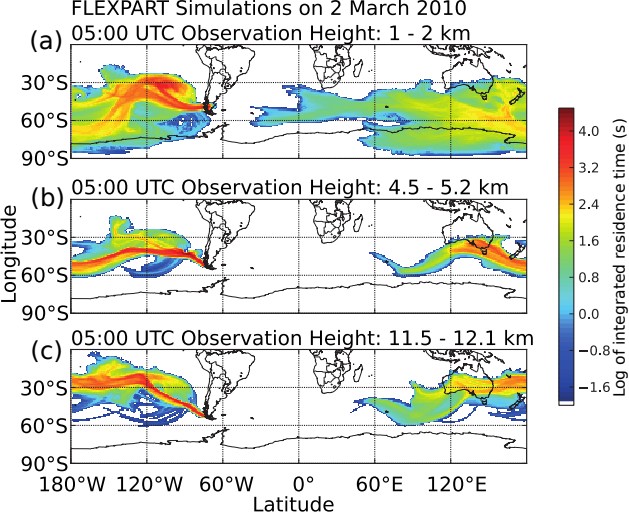

**Figure 3.** FLEXPART simulations for the integrated residence time of the particles that traveled in the whole atmospheric column within the last ten days until the observation time on 2 March 2010 for the observed heights between 1 and 2 km (a), 4.5 and 5.2 km (b) and 11.5 and 12.1 km (c). The colors represent the logarithm of the integrated residence time (in seconds) in a grid box for 10–day integration time.

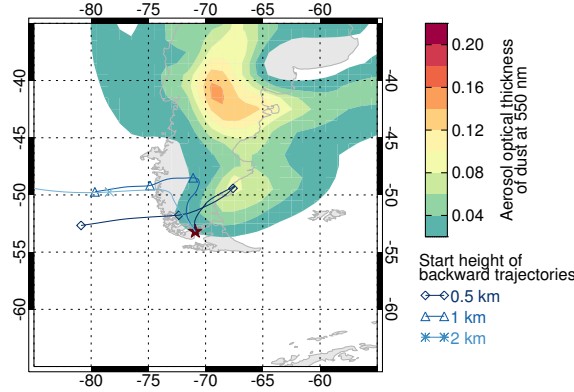

**Figure 4.** HYSPLIT 48–hours backward trajectories for heights of 0.5, 1 and 2 km for Punta Arenas on 2 March 2010, 09:00 UTC and modelled dust surface concentrations from the NAAPS aerosol model on 2 March 2010, 06:00 UTC.





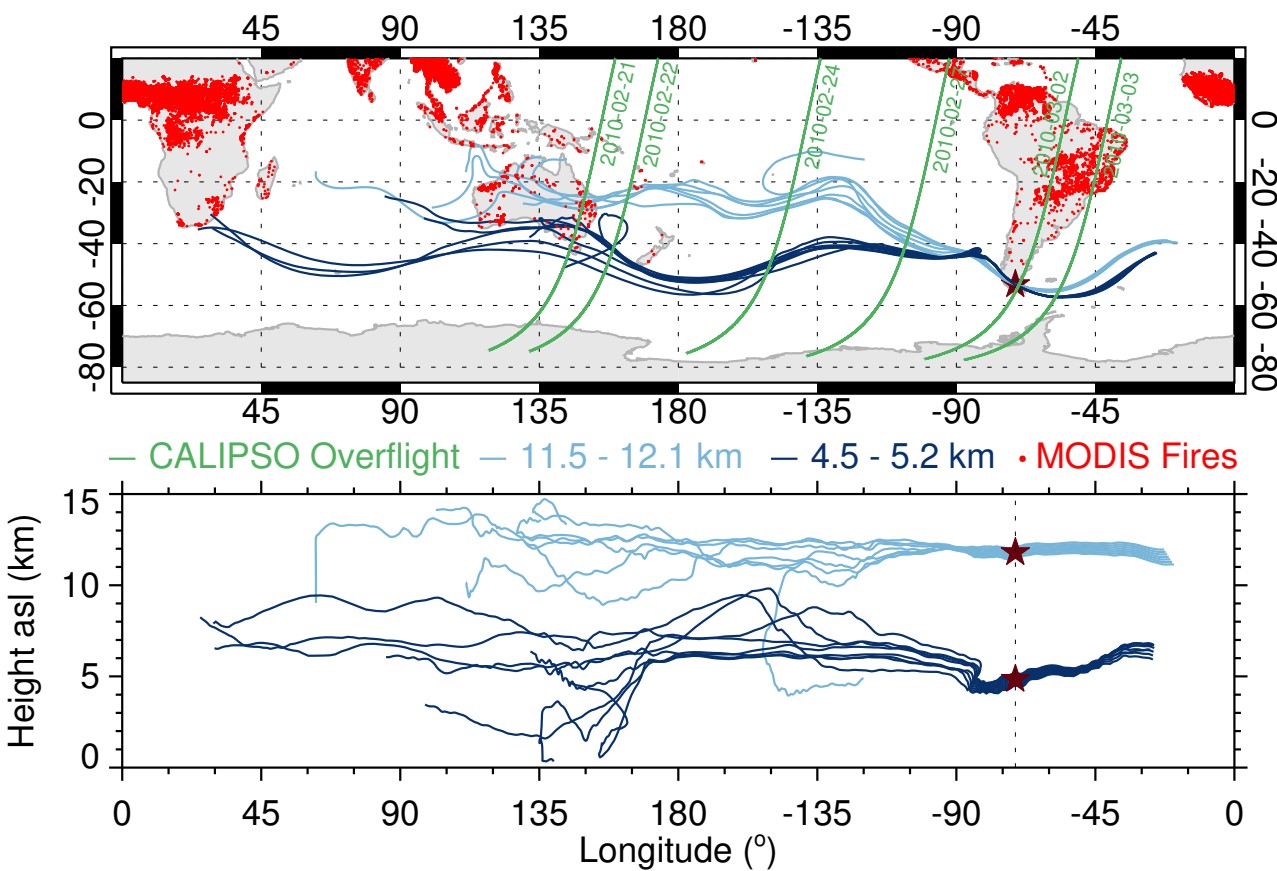

**Figure 5.** Top: Map of the HYSPLIT 13–days backward trajectories for heights of 4.5 to 5.2 km (blue) and 11.5 to 12.1 km (light blue) arriving at Punta Arenas (brown star) on 2 March 2010, 09:00 UTC, MODIS fire counts between 17 to 25 February 2010 (red dots) and CALIPSO tracks (green). Bottom: Height of the according trajectories.



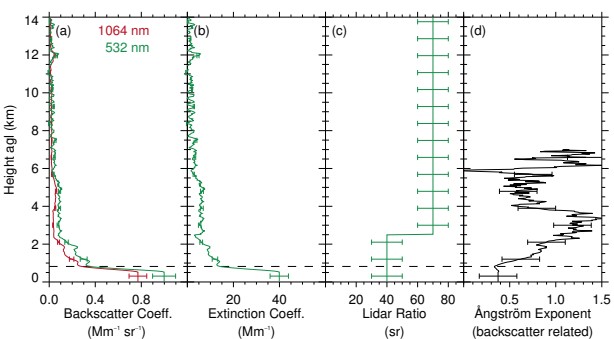

**Figure 6.** Vertical profiles of (a) particle backscatter coefficient at 532 nm and 1064 nm with error bars indicating 10 % uncertainty, (b) particle extinction coefficient at 532 nm with error bars resulting from a lidar ratio uncertainty of $\pm 10$ sr, (c) particle lidar ratio with error bars of $\pm 10$ sr and (d) backscatter related Ångström exponent with propagated error bars, derived by Polly[XT] on 2 March 2010, 06:15 to 09:00 UTC. The planetary boundary layer top height is indicated by the black dashed line.





**Figure 7.** Height-time display of the total attenuated backscatter at 532 nm (left panel) and corresponding aerosol subtype (right panel) derived by CALIOP from six CALIPSO overpasses on (a,b) 21, (c,d) 22, (e,f) 24, (g,h) 27 February, (i,j) 2 and (k,l) 3 March 2010. The aerosol subtypes are only illustrated if the CAD score is below -80 excluding uncertain classifications.





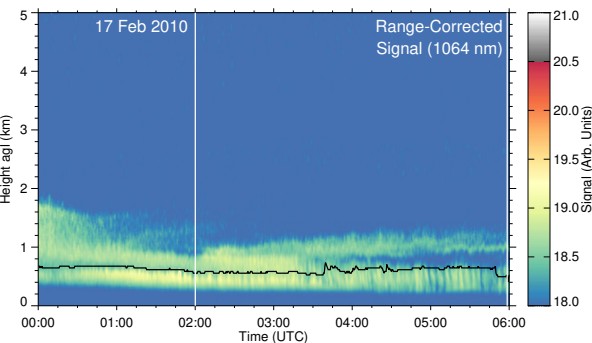

**Figure 8.** Height–time display of the range–corrected signal observed at 1064 nm with the Polly[XT] lidar on 17 February 2010. The analysed period between 02:00 bis 06:00 UTC is framed white. The planetary boundary layer top height is indicated by the black line.

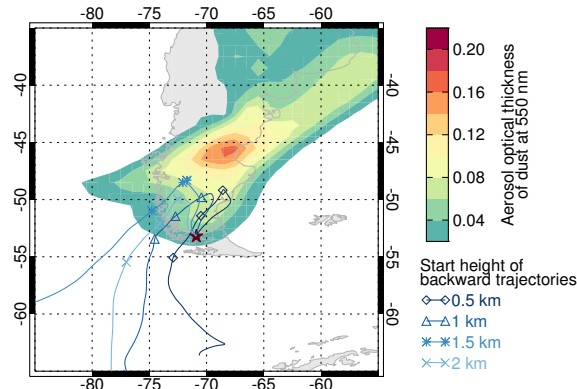

**Figure 9.** HYSPLIT 96–hours backward trajectories for heights of 0.5, 1, 1.5 and 2 km for Punta Arenas on 17 February 2010, 03:00 UTC and modelled dust surface concentrations from the NAAPS aerosol model on 17 February 2010, 06:00 UTC.





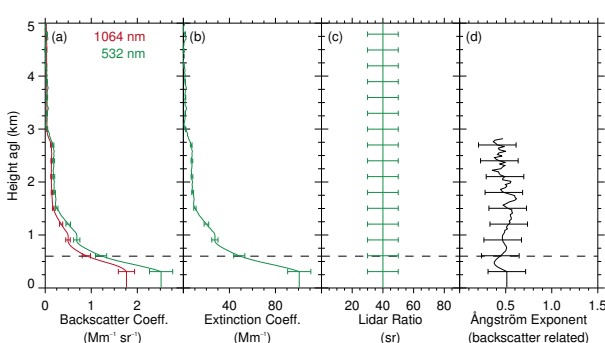

**Figure 10.** Vertical profiles of (a) particle backscatter coefficient at 532 nm and 1064 nm, with error bars indicating 10 % uncertainty, (b) particle extinction coefficient at 532 nm, with error bars resulting from a lidar ratio uncertainty of $\pm 10$ sr, (c) particle lidar ratio with error bars of $\pm 10$ sr and (d) backscatter related Ångström exponent with propagated error bars, derived by Polly$^{\mathrm{XT}}$ on 17 February 2010, 02:00 to 06:00 UTC. The planetary boundary layer top height is indicated by the black dashed line.





**Figure 11.** (a) Monthly average of CALIOP Level 3 AOT (at 532 nm) and their standard deviation at Punta Arenas (blue) and Rio Gallegos (red) in 2009 and 2010. The ALPACA campaign is indicated by the shaded area. (b) Daily average of AERONET Sun photometer AOT (at 500 nm) at Rio Gallegos for the ALPACA campaign. (c) Height-time display of the range–corrected signal (at 1064 nm) measured by Polly[XT]. Panel d) shows the accumulated residence time of backward trajectories separated by different regions of origin.



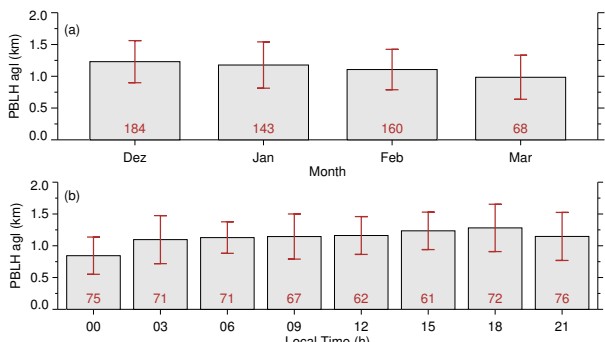

**Figure 12.** Height of the PBL determined by Polly$^{XT}$ between 4 Dezember 2009 and 31 March 2010. a) Monthly averaged PBL top heights including the standard deviation as error bars and b) averaged PBL top height as function of time of day (averaged over three hours) including standard deviation as error bars. The red numbers indicate the number of measurements of the according bar. Local Time is UTC −4 hours.

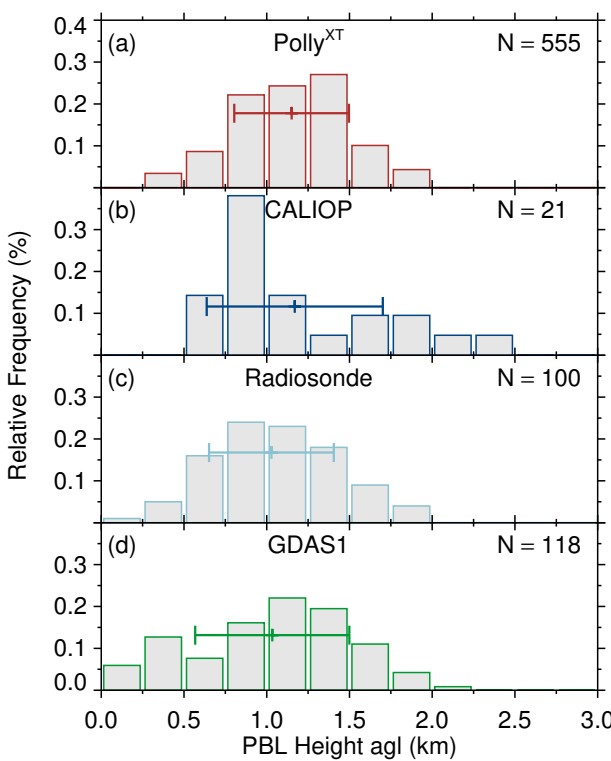

**Figure 13.** Frequency distribution of the PBL heights determined by a) Polly$^{XT}$ between 4 Dezember 2009 and 31 March 2010., b) CALIOP between 1 May 2009 and 30 April 2010, c) radiosonde (12 UTC) and d) GDAS1 data (12 UTC) from 4 December 2009 to 31 March 2010 with an increment of 250 m, respectively. N gives the number of samples.





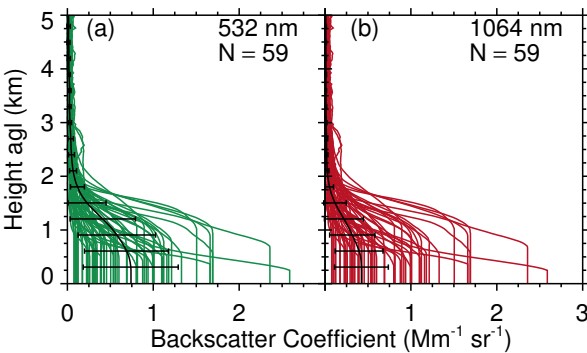

**Figure 14.** Single (coloured lines) and averaged (black lines) height profiles of the particle backscatter coefficients at 532 nm (a) and at 1064 nm (b) as derived from Polly$^{XT}$ observations. The error bars indicate the standard deviations.

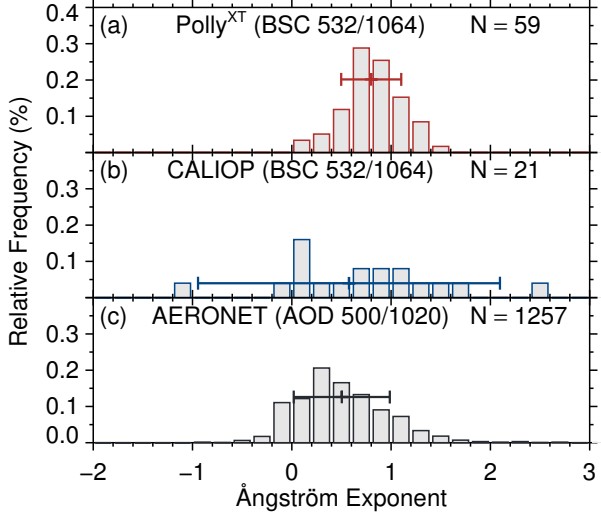

**Figure 15.** Frequency distribution of the vertical integrated particle–backscatter–related Ångström exponent (532 and 1064 nm) of a) Polly$^{XT}$ and b) CALIOP with an increment of 0.2. Additionally, the frequency distribution of the AOT–related Ångström exponent determined by Sun photometer measurements in Rio Gallegos is illustrated. N gives the number of measurements.