# Peer review of "Vertical aerosol distribution in the Southern hemispheric Midlatitudes as observed with lidar at Punta Arenas, Chile (53.2°S and 70.9°W) during ALPACA."

_Atmospheric Chemistry and Physics, 2018_

## Referee Comment (RC1) · Anonymous Referee #1 · 7 Jan 2019

This paper presents the first results regarding aerosol research in an almost pristine atmosphere over a Southern hemisphere site. Due to the lack of knowledge in such regions, this kind of papers are relevant, event if the involved databases are not large enough. The authors present their study in a proper, robust way. The topic fits in this journal, but my recommendation is to improve the aspects listed as follows before publication:

Page 2, lines 21-22: Consider to review all the lidar-related papers regarding aerosol and cloud profiling from ground. I suggest to start with LALINET (lalinet.org), and then

[Figure]

you should review other papers developed by indidivual stations and/or during special campaign. Here some of them are listed:

Carmen Córdoba-Jabonero, Fabio J.S.Lopes, Eduardo Landulfo, Emilio Cuevas, Héctor Ochoa, Manuel Gil-Ojeda, Diversity on subtropical and polar cirrus clouds properties as derived from both ground-based lidars and CALIPSO/CALIOP measurements, Atmospheric Research Volume 183, 1 January 2017, Pages 151-165.

Gouveia, D. A., Barja, B., Barbosa, H. M. J., Seifert, P., Baars, H., Pauliquevis, T., and Artaxo, P.: Optical and geometrical properties of cirrus clouds in Amazonia derived from 1 year of ground-based lidar measurements, Atmos. Chem. Phys., 17, 3619-3636, https://doi.org/10.5194/acp-17-3619-2017, 2017.

Guerrero-Rascado, J. L. et al. (2014). Multispectral elastic scanning lidar for flares research: characterizing the electronic subsystem and application Optics Express. OSA. 22-25, pp.31063-31077. ISSN 1094-4087

Lopes, F. J. S., Landulfo, E., and Vaughan, M. A.: Evaluating CALIPSO's 532 nm lidar ratio selection algorithm using AERONET sun photometers in Brazil, Atmos. Meas. Tech., 6, 3281-3299, https://doi.org/10.5194/amt-6-3281-2013, 2013.

Page 2, line 32: replace "lidarÂ" by "lidar".

Page 2, line 33: replace "ALAPACA" by "ALPACA".

Page 3, lines 24-26: Why is it necessary to do this strong assumption? The instrument used here is a Raman lidar system and, therefore, it is possible to apply the method developed in Wandinger and Ansmann (2002) to derive a proper experimental overlap function to correct for this effect.

Wandinger U., Ansmann A., Experimental determination of the lidar overlap profile with Raman lidar, Appl Opt. 2002 Jan 20;41(3):511-4.

Page 3, lines 27-28: The Wavelet method is known to provide reliable results under
scenarios with absence of lofted aerosol particles in the free troposphere of decoupled plumes above a well mixed PBL. However, as several works pointed out, such as Granados-Muñoz et al (2012), this method cannot unambiguously identify the PBL top in a stratified PBL. Please consider to apply another more recent approaches, some of them listed below:

Bravo-Aranda, J. A., de Arruda Moreira, G., Navas-Guzmán, F., Granados-Muñoz, M. J., Guerrero-Rascado, J. L., Pozo-Vázquez, D., Arbizu-Barrena, C., Olmo Reyes, F. J., Mallet, M., and Alados Arboledas, L.: A new methodology for PBL height estimations based on lidar depolarization measurements: analysis and comparison against MWR and WRF model-based results, Atmos. Chem. Phys., 17, 6839-6851, https://doi.org/10.5194/acp-17-6839-2017, 2017.

Geiß, A., Wiegner, M., Bonn, B., Schäfer, K., Forkel, R., von Schneidemesser, E., Münkel, C., Chan, K. L., and Nothard, R.: Mixing layer height as an indicator for urban air quality?, Atmos. Meas. Tech., 10, 2969-2988, https://doi.org/10.5194/amt-10-2969-2017, 2017.

Poltera, Y., Martucci, G., Collaud Coen, M., Hervo, M., Emmenegger, L., Henne, S., Brunner, D., and Haefele, A.: PathfinderTURB: an automatic boundary layer algorithm. Development, validation and application to study the impact on in situ measurements at the Jungfraujoch, Atmos. Chem. Phys., 17, 10051-10070, https://doi.org/10.5194/acp-17-10051-2017, 2017.

Pal, S., Haeffelin, M., & Batchvarova, E. (2013). Exploring a geophysical process-based attribution technique for the determination of the atmospheric boundary layer depth using aerosol lidar and near-surface meteorological measurements. Journal of Geophysical Research: Atmospheres, 118(16), 9277-9295.

Granados-Muñoz, M. J., Navas-Guzmán, F., Bravo-Aranda, J. A., Guerrero-Rascado, J. L., Lyamani, H., Fernández-Gálvez, J., Alados-Arboledas, L. (2012). Automatic determination of the planetary boundary layer height using lidar: One-year analysis over

southeastern Spain. Journal of Geophysical Research: Atmospheres, 117(D18).

Page 3, lines 29-31: Authors say that the low signal-to-noise ratio of the Raman signals prevented the determination independent particle extinction coefficients. I wonder if they try to increase the spatial smooth and/or the integration time.

Page 4, lines 21-25: Why is it necessary to use two different models for air mass transport analysis? What is the actual advantage of using both?

Page 4, line 32: Be carefull with the use of mixing depth. I guess that the authors refers to PBL top height. Mixing depth refers to PBL top height only under convective scenarios. In my opinion this ambiguous term must be replaced. Moreover, I'm curious about how this values is computed from the GDAS data.

Page 5, lines 5-6: Please, consider to add information on other models such as CAMS (https://atmosphere.copernicus.eu/charts/cams/aerosol-forecasts?facets=undefined&time=2019010500,3,2019010503&projection=classical_global&layer_name=composition_ao and/or NMMB/BSC-Dust model (https://dust.aemet.es/methods/the-nmmb-bsc-dust-model)

Page 5, lines 23-25: Uncertainties in determing backtrajectories increase with duration of the computed backtrajectories. In my opinion trajectories with a duration as high as 13 days have large uncertainties. How can you guarantee that the computed backtrajectories overpassed regions with active fires?

Page 6, lines 1-3: Did the mentioned particle lidar ratio values apply to both 532 and 1064 nm? Provide explicitaly this information.

Page 6, lines 5-11: Were thess Angstrom exponents computed over backscatter or over extinction? Please, especify it.

Page 7, line 1, and page 8, lines 1-5: Especify in the text the wavelength for these AOT values.

Page 7, line 30: Provide the distance between Punta Arenas and Rio Gallegos.

Page 8, line 24: To be comparable with previous syudies, I recommend to perform the analysis on 1h basis.

Page 8, lines 26-27: I lot of studies on PBL height have been conducted in the framework of EARLINET. Please, include some of them, in particular with similar latitude and another sites with similar distance to ocean/sea.

Page 8, line 34: Include distance to the lidar site.

Page 9, lines 21-22: Consider my previous comment on the overlap function.

Page 9, lines 31-32: I don't understand the concept "integrated backscatter-related Angström Exponent". Backscatter-related Angström exponent is an intensive property, therefore its integration doesn't make sense. Maybe you were referring to vertically averaged backscatter-related Angström exponent.

Figure 11: why do the standard deviation for Rio Gallegos is too high in March 2009 and July 2010 (Fig. 11a)?. I recommend to include labels on x-axis in Figure 11b and 11c, not only in figure 11d.

Figure 13 and 15: what is the horizontal error bar?

---

## Referee Comment (RC2) · Anonymous Referee #2 · 16 Jan 2019

The authors present continuous Raman Lidar measurements of the aerosol backscatter and extinction from Punta Arenas, Chile that cover four months of the southern summer of 2009-2010. Data from a nearby AERONET site, and the CALIOP remote lidar are used to provide context. From this dataset the authors identify eight periods with aerosol layers aloft. Two such cases are presented in detail with back trajectories indicating the sources of primary particles as Australian biomass burning in one case and Patagonian dust in the other. The authors also report the average aerosol profile for the whole measurement period, relate it to the back trajectories, and compare the

[Figure]

PBL from lidar profiles and a few other sources. Generally, the aerosol concentrations are low and are near the detection limit of the lidar in free troposphere.

This manuscript mainly reports the measurements and performs basic analyses and seems to be written in anticipation of future measurements. The authors confirm the conclusion of other studies that aerosol over southern Chile is representative of "pristine, pre-industrial" conditions. The authors could do a better in connecting the analyses to this conclusion. How do the case studies affect this conclusion? Why is PBL height discussed in detail and compared from four different measurements? The manuscript has several figure that are only briefly described in the text. Are all the figures necessary?

Specific Comments: P3L1: There were eight cases of aerosol layers aloft but only 2 cases were presented. Could you add sentence or two describing the other cases (similar? Closer to detection limit?) and why they were not presented?

P3L29: I'm confused because the Raman signal is from molecular backscatter and not related to the aerosol content.

P4L7: This sentence is confusing because is switches between level 3 and version 4.10.

P4L19: Could you add the distance between the two sites?

P5 L19: replace "locally exceeded 0.14 and 0.02 in the area of Punta" with "regionally peaked at 0.14 and was 0.02 in the area of Punta"

P6L21: This comparison is not very useful in that the AOT from a biomass burning event is an extensive property depends on the size and fire intensity and its subsequent dispersion in the atmosphere. Why are two very different events being compared? P7L28: 'numbers' should be singular

P9L13-15: I infer this conclusion is that Polly and CALIOP PBL height are determined by a decrease in aerosol backscatter, and the radiosonde and GDAS1 PBL heights

are determined by the potential temperature profile. The fact that they agree mean the aerosol top of the aerosol layer coincides with the temperature inflection; hence the aerosol is in the PBL. The author may want to state this reasoning more explicitly rather than having the reader infer it.
* * *

---

## Author Comment (AC1) · 21 Mar 2019

**Response to Reviewers #1 and #2**

**We like to thank the reviewers for providing helpful comments to improve the manuscript. All changes are highlighted in the manuscript file. Added text is wavy-underlined and blue, discarded text is struck out and red.**
**Additionally, there is a new contributor stated as coauthor, Michael Fromm.**

**Response to Reviewer #1**

a) General comments:

This paper presents the first results regarding aerosol research in an almost pristine atmosphere over a Southern hemisphere site. Due to the lack of knowledge in such regions, this kind of papers are relevant, event if the involved databases are not large enough. The authors present their study in a proper, robust way. The topic fits in this journal, but my recommendation is to improve the aspects listed as follows before publication:

b) Detailed comments:

Page 2, lines 21-22: Consider to review all the lidar-related papers regarding aerosol and cloud profiling from ground. I suggest to start with LALINET (lalinet.org), and then you should review other papers developed by individual stations and/or during special campaign.
**We added two of the suggested publications. The LALINET publications are cited in the next paragraph where we introduce networks as EARLINET and LALINET.**

Page 2, line 32: replace "lidarÂ" by "lidar".
**Done as suggested.**

Page 2, line 33: replace "ALAPACA" by "ALPACA".
**Done as suggested.**

Page 3, lines 24-26: Why is it necessary to do this strong assumption? The instrument used here is a Raman lidar system and, therefore, it is possible to apply the method developed in Wandinger and Ansmann (2002) to derive a proper experimental overlap function to correct for this effect.
**We corrected the confusing text:**
**In fact, below 1500 m the laser beam with the receiver field of view of the bistatic system is incomplete, thus an overlap correction was applied. However, below 400 m the overlap function is less than 0.5 and a reliable overlap correction is not possible. As a consequence, values of the particle backscatter coefficient were set constant below 400 m height, in the lower part of the planetary boundary layer (PBL) under the assumption of well-mixed conditions.**

Page 3, lines 27-28: The Wavelet method is known to provide reliable results under scenarios with absence of lofted aerosol particles in the free troposphere of decoupled plumes above a well mixed PBL. However, as several works pointed out, such as Granados-Muñoz et al. (2012), this method cannot unambiguously identify the PBL top in a stratified PBL. Please consider to apply another more recent approaches, some of them listed below:
**Due to the strong wind occurring in Punta Arenas a stable nighttime layering as in Europe is not observed there. Thus, potential misclassification due to stratification during nighttime are rare. Nevertheless, visual inspection to all derived PBL-tops was performed to ensure no misclassification.**

Page 3, lines 29-31: Authors say that the low signal-to-noise ratio of the Raman signals prevented the determination independent particle extinction coefficients. I wonder if they try to increase the spatial smooth and/or the integration time.
**We already optimized the spatial and temporal resolution for the optical aerosol profiles, but considering the prerequisite of having a temporal nearly homogeneous layering it was impossible to further increase averaging periods.**

Page 4, lines 21-25: Why is it necessary to use two different models for air mass transport analysis? What is the actual advantage of using both?
**Using two different models makes the results more robust. HYSPLIT runs for specific trajectories are much faster than the FLEXPART analyses which performs calculation for a large number of trajectories.**

Page 4, line 32: Be careful with the use of mixing depth. I guess that the authors refers to PBL top height. Mixing depth refers to PBL top height only under convective scenarios. In my opinion this ambiguous term must be replaced. Moreover, I'm curious about how this values is computed from the GDAS data.
**We changed the notation from mixing depth to PBL height.**
**The PBL height is determined according to:**
**Troen, I.B. & Mahrt, L. Boundary-Layer Meteorol (1986) 37: 129. https://doi.org/10.1007/BF0012276 (https://link.springer.com/article/10.1007/BF00122760).**
**It is a standard method from the GFS model based on virtual potential temperature and wind vector.**

Page 5, lines 5-6: Please, consider to add information on other models such as CAMS (https://atmosphere.copernicus.eu/charts/cams/aerosol-forecasts facets=undefined&time=2019010500,3,2019010503&projection=classical_global&layer_name=compositio and/or NMMB/BSC-Dust model (https://dust.aemet.es/methods/the-nmmb-bsc-dust-model)
**We added the information of the MACC aerosol optical thickness of dust in the data description as well as at both case studies.**

Page 5, lines 23-25: Uncertainties in determing backtrajectories increase with duration of the computed backtrajectories. In my opinion trajectories with a duration as high as 13 days have large uncertainties. How can you guarantee that the computed backtrajectories overpassed regions with active fires?
**We agree that the uncertainties of the HYSPLIT trajectories increase with duration. For that reason we additionally used the FLEXPART simulation which provide the residence time of air parcels in a specific area computed by a large amount of trajectories.**
**HYSPLIT uncertainties are explained in the manual (https://www.arl.noaa.gov/documents/workshop/NAQC2007/HTML_Docs/trajerro.html) :**
**'Overall, from the literature, one can estimate the total error to be anywhere from 15 to 30% of the travel distance.'**

Page 6, lines 1-3: Did the mentioned particle lidar ratio values apply to both 532 and 1064 nm? Provide explicitly this information.
**The mentioned particle lidar ratio corresponds to the 532 nm channel because there is no investigation of extinction coefficients at 1064 nm. The missing information is added as suggested.**

Page 6, lines 5-11: Were these Angstrom exponents computed over backscatter or over extinction? Please, especify it.
**These values are backscatter-related. Specified as suggested.**

Page 7, line 1, and page 8, lines 1-5: Especify in the text the wavelength for these AOT values.
**Missing wavelength information added as suggested.**

Page 7, line 30: Provide the distance between Punta Arenas and Rio Gallegos.
**Information is added in the beginning of section 3.3, auxiliary data.**

Page 8, line 24: To be comparable with previous studies, I recommend to perform the analysis on 1h basis.
**We performed this analysis on a three hour basis to ensure a homogeneous distribution of samples in each bin.**

Page 8, lines 26-27: A lot of studies on PBL height have been conducted in the framework of EARLINET. Please, include some of them, in particular with similar latitude and another sites with similar distance to ocean/sea.
**We added Granada, Spain, as site with similar distance to the coast and mentioned that Leipzig has a comparable latitude as Punta Arenas.**

Page 8, line 34: Include distance to the lidar site.
**Added as suggested.**

Page 9, lines 21-22: Consider my previous comment on the overlap function.
**Explained above in the previous comment.**

Page 9, lines 31-32: I don't understand the concept "integrated backscatter-related Angström Exponent". Backscatter-related Angström exponent is an intensive property, therefore its integration doesn't make sense. Maybe you were referring to vertically averaged backscatter-related Angström exponent.
**That is right. We refer to vertically averaged values. Changed as suggested.**

Figure 11: why do the standard deviation for Rio Gallegos is too high in March 2009 and July 2010 (Fig. 11a)?
**The high value could be caused by a low sample size with a high variation between the samples or caused by outliers. This is not reported in the CALIPSO L3 data.**

I recommend to include labels on x-axis in Figure 11b and 11c, not only in figure 11d.
**We tried, but it was confusing. The clear arrangement of the figure would be downgraded.**

Figure 13 and 15: what is the horizontal error bar?
**It indicates the standard deviation. Added.**

**Response to Reviewer #2**

a) General comments:

The authors present continuous Raman Lidar measurements of the aerosol backscatter and extinction from Punta Arenas, Chile that cover four months of the southern summer of 2009-2010. Data from a nearby AERONET site, and the CALIOP remote lidar are used to provide context. From this dataset the authors identify eight periods with aerosol layers aloft. Two such cases are presented in detail with back trajectories indicating the sources of primary particles as Australian biomass burning in one case and Patagonian dust in the other. The authors also report the average aerosol profile for the whole measurement period, relate it to the back trajectories, and compare the PBL from lidar profiles and a few other sources. Generally, the aerosol concentrations are low and are near the detection limit of the lidar in free troposphere.

This manuscript mainly reports the measurements and performs basic analyses and seems to be written in anticipation of future measurements. The authors confirm the conclusion of other studies that aerosol over

southern Chile is representative of "pristine, pre-industrial" conditions. The authors could do a better in connecting the analyses to this conclusion. How do the case studies affect this conclusion?
**We improved this part according to your suggestions. Now our results are more connected to the conclusions of Hamilton et al. (2014) and Carslaw et al. (2017).**

Why is PBL height discussed in detail and compared from four different measurements?
**The PBL height is discussed very detailed because most aerosol load is within the pbl. A comprehensive characterization of the pbl height makes the results more robust and certain.**

The manuscript has several figure that are only briefly described in the text. Are all the figures necessary?
**The authors agree that there are many figures, but all of them help the reader to get the message of the studies and give interesting information to the community.**

b) Detailed comments:

Specific Comments: P3L1: There were eight cases of aerosol layers aloft but only 2 cases were presented. Could you add sentence or two describing the other cases (similar? Closer to detection limit?) and why they were not presented?
**The other six cases with lofted aerosol layers are not presented because they are similar to those presented here. We added this information to the text.**

P3L29: I'm confused because the Raman signal is from molecular backscatter and not related to the aerosol content.
**We discarded the last part of the sentence.**

P4L7: This sentence is confusing because is switches between level 3 and version 4.10.
**Corrected.**

P4L19: Could you add the distance between the two sites?
**Done as suggested.**

P5L19: replace "locally exceeded 0.14 and 0.02 in the area of Punta" with "regionally peaked at 0.14 and was 0.02 in the area of Punta"
**Done as suggested.**

P6L21: This comparison is not very useful in that the AOT from a biomass burning event is an extensive property depends on the size and fire intensity and its subsequent dispersion in the atmosphere. Why are two very different events being compared?
**We mention the AOT from an Amazonian site to relate the values to extreme values to get an impression about the spread.**

P7L28: 'numbers' should be singular
**Done as suggested.**

P9L13-15: I infer this conclusion is that Polly and CALIOP PBL height are determined by a decrease in aerosol backscatter, and the radiosonde and GDAS1 PBL heights are determined by the potential temperature profile. The fact that they agree mean the aerosol top of the aerosol layer coincides with the temperature inflection; hence the aerosol is in the PBL. The author may want to state this reasoning more explicitly rather than having the reader infer it.
**Done as suggested**

[revised manuscript text omitted]